# Unlocking the Potential of Human-Induced Pluripotent Stem Cells: Cellular Responses and Secretome Profiles in Peptide Hydrogel 3D Culture

**DOI:** 10.3390/cells13020143

**Published:** 2024-01-12

**Authors:** Muyun Cui, Wei Wu, Quan Li, Guangyan Qi, Xuming Liu, Jianfa Bai, Mingshun Chen, Ping Li, Xiuzhi (Susan) Sun

**Affiliations:** 1Department of Grain Science and Industry, Kansas State University, Manhattan, KS 66506, USA; muyunc@ksu.edu (M.C.); guangyan@ksu.edu (G.Q.); 2Department of Chemistry, Kansas State University, Manhattan, KS 66506, USA; gisswu@ksu.edu (W.W.); pli@ksu.edu (P.L.); 3Carl and Melinda Helwig Department of Biological and Agricultural Engineering, Kansas State University, Manhattan, KS 66506, USA; quanl@ksu.edu; 4USDA-ARS and Department of Entomology, Kansas State University, Manhattan, KS 66506, USA; xmliu@ksu.edu (X.L.); mchen@ksu.edu (M.C.); 5Department of Diagnostic Medicine/Pathobiology, College of Veterinary Medicine, Kansas State University, Manhattan, KS 66506, USA; jbai@vet.ksu.edu

**Keywords:** hiPSCs, 3D culture, spheroids, PGmatrix, hydrogel, extracellular vesicles, secretome

## Abstract

Human-induced pluripotent stem cells (hiPSCs) have shown great potential for human health, but their growth and properties have been significantly limited by the traditional monolayer (2D) cell culture method for more than 15 years. Three-dimensional (3D) culture technology has demonstrated tremendous advantages over 2D. In particular, the 3D PGmatrix hiPSC derived from a peptide hydrogel offers a breakthrough pathway for the maintenance and expansion of physiologically relevant hiPSC 3D colonies (spheroids). In this study, the impact of 3D culture conditions in PGmatrix hiPSC on cell performance, integrity, and secretome profiles was determined across two commonly used hiPSC cell lines derived from fibroblast cells (hiPSC-F) and peripheral blood mononuclear cells (hiPSC-P) in the two most popular hiPSC culture media (mTeSR1 and essential eight (E8)). The 3D culture conditions varied in hydrogel strength, 3D embedded matrix, and 3D suspension matrix. The results showed that hiPSCs cultured in 3D PGmatrix hiPSC demonstrated the ability to maintain a consistently high cell viability that was above 95% across all the 3D conditions with cell expansion rates of 10–20-fold, depending on the 3D conditions and cell lines. The RT-qPCR analysis suggested that pluripotent gene markers are stable and not significantly affected by the cell lines or 3D PGmatrix conditions tested in this study. Mass spectrometry-based analysis of secretome from hiPSCs cultured in 3D PGmatrix hiPSC revealed a significantly higher quantity of unique proteins, including extracellular vesicle (EV)-related proteins and growth factors, compared to those in the 2D culture. Moreover, this is the first evidence to identify that hiPSCs in a medium with a rich supplement (i.e., mTeSR1) released more growth-regulating factors, while in a medium with fewer supplements (i.e., E8) hiPSCs secreted more survival growth factors and extracellular proteins. These findings offer insights into how these differences may impact hiPSC behavior, and they deepen our understanding of how hiPSCs respond to 3D culture conditions, aiding the optimization of hiPSC properties in translational biomedical research toward clinical applications.

## 1. Introduction

Human-induced pluripotent stem cells (hiPSCs) have emerged as highly promising and versatile tools across a multitude of domains including disease modeling [1], cell therapy [2], drug discovery [3,4], and toxicology screening [5,6]. Their applications also span a wide range of disciplines, encompassing fields such as neurodegeneration, hepatology, hematology, immunology, and cardiology [7]. The distinctive advantages of hiPSCs, that stem from their differentiation capacity, allogeneic nature, abundance in source, and broad clinical applicability [8], have driven a rapid evolution in their cultivation techniques to meet the diverse demands of advanced research and clinical applications. Of particular significance is the evolution of hiPSC culture within a three-dimensional (3D) context. The adoption of 3D culture methods has opened up new avenues for hiPSC investigation and application, owing to the culture’s ability to mimic the in vivo cellular structure and conditions [9,10]. Innovative 3D cultivation in a peptide hydrogel (PepGel PGmatrix hiPSC) has enabled the production of physiologically relevant spheroids and organoids [11,12]. Recently developed 3D technologies include polyethylene glycol (PEG)-based hydrogels and adult hiPSC-aggregated spheroids, which use non-adherent well plates or microplates induced by mechanical molding forces. This high content and physiologically relevant PGmatrix hiPSC technology enhances the accuracy of disease modeling and dynamics, and lays the foundation for tissue and organ regeneration [1,11]. Additionally, the convergence of 3D culture techniques with cutting-edge technologies including 3D printing has unlocked unprecedented possibilities in regenerative medicine and beyond [11,13]. Furthermore, this PGmatrix hiPSC is built on a tri-block synthetic peptide [14,15]. This peptide hydrogel not only promotes biocompatibility and presents no immunogenic effects [16,17], but its unique properties have also proven beneficial for various applications: 3D culture of various cancer cells [12,18,19,20,21]; 3D DNA transfection [22]; and in vivo delivery of cells, drugs, antigens, and viruses [16,23].

The dynamic nature of hiPSC applications has necessitated an agile adaptation of culture methodologies to cater to these burgeoning needs. This research endeavors to delve into the intricate interplay between culture conditions with PGmatrix hiPSC and cellular performance within the realm of hiPSCs. The primary focus is on the hiPSCs derived from peripheral blood mononuclear cells (PBMCs), a choice driven by the distinct advantage of acquiring PBMCs through minimally invasive procedures, in contrast to the more intrusive surgical operations required for obtaining fibroblasts. The two selected cell lines exhibit distinctive attributes that are inherited from their respective parental cells, and their cellular activities are further influenced by epigenetic memory. The evaluation centers on two widely used culture media, namely mTeSR and essential eight (E8) for hiPSCs. However, the experimental framework extends further to encompass diverse variables, including a spectrum of hydrogel concentrations and a comparison between 3D embedded and 3D suspension matrix conditions. These variables collectively contribute to a comprehensive examination of the impact of culture conditions on hiPSC performance and properties. 

Employing a multifaceted approach, besides hiPSC growth performance, the research employs gene expression analysis, protein profiling, and quantification of growth factor and extracellular vesicle (EV) release. This approach seeks to unravel the intricate relationships that govern hiPSC responses within the context of 3D culture environments. By shedding light on the nuanced interactions between culture conditions and cellular outcomes, the study not only enhances our understanding of hiPSC behavior but also opens doors to pioneering advancements in the field. This investigation marks a significant step toward harnessing the potential of hiPSCs within 3D culture systems, promising improved physiological relevance and the potential for groundbreaking discoveries. As research continues to unlock the full potential of hiPSCs, their application in various domains holds the promise of transforming approaches to disease understanding, therapeutic interventions, and drug development. This paradigm shift signifies not only the progression of hiPSC-based research but also holds the potential to revolutionize our approach to addressing complex medical challenges.

## 2. Method

### 2.1. Materials and 3D Cell Culture Conditions

The PGmatrix hiPSC matrix (PepGel LLC, Manhattan, KS, USA) was used for 3D embedded hiPSC cultures. The PGmatrix 3D Suspension hiPSC matrix (PepGel LLC, Manhattan, KS, USA) was used for 3D suspension culture of hiPSCs. hiPSCs (p9–p11) derived from peripheral blood mononuclear cells (PBMCs) (Applied StemCell Inc., Milpitas, CA, USA) were used in this research, and hiPSCs (p9–p11) derived from fibroblasts (Applied StemCell) were used for comparison. The hiPSCs (p9–p11) were cultured in 3D for three passages before official experiments started. The culture medium mTeSR1 and its supplements were products from StemCell Technology (Seattle, WA, USA) and E8 was from ThermoFisher (Chicago, IL, USA). The 3D culture conditions and short names used in Figure labels are shown in Table 1.

### 2.2. hiPSC 3D Embedded Culture within PGmatrix hiPSC

The manufacturer’s user guide for the PGmatrix hiPSC kit was followed. Briefly, 1 × 10^5^ cells/mL hiPSC cell suspension in complete mTeSR1 was first mixed with PGworks (PepGel LLC, Manhattan, KS, USA), then PGmatrix hiPSC (PG) was added to the mixture and mixed thoroughly, to reach final concentrations of 0.2%, 0.5%, and 1%. The complete mTeSR1 medium was prepared by adding mTeSR1 supplements and PGgrow (PepGel LLC, Manhattan, KS, USA) at a ratio of 1000:1 (*v*/*v*, medium: PGgrow). The complete E8 medium was prepared by adding PGgrow at a ratio of 1000:1 (*v*/*v*, medium: PGgrow). The mixing ratio of cell suspension, PGworks, and PG (*v*/*v*/*v*) followed the PGmatrix hiPSC user’s guide (PepGel LLC, Manhattan, KS, USA). The mixture with a cell density of 2 × 10^5^ cell/mL was plated in a 24-well plate at 500 µL/well and incubated at 37 °C for 30 min to form a hydrogel. After gelation, the complete mTeSR1 or E8 media was added gently on top of the hydrogel to feed the cells, and the complete culture medium was changed on day 3 and day 4. hiPSC spheroids were harvested on day 5 and dissociated for cell viability and proliferation analysis. The cell suspension was then used for the next passage or other experiments and analysis. 

### 2.3. hiPSC 3D Suspension Culture within the PGmatrix 3D Suspension Matrix

The manufacturer’s user guide for the PGmatrix 3D Suspension hiPSC kit was followed. Briefly, the hiPSC cell suspension in complete mTeSR1 medium was mixed thoroughly with the PGmatrix 3D Suspension hiPSC matrix (PG Susp) solution and PGworks at ratios of 2:1:0.03 (cell suspension: PG Susp: PGworks), to achieve a cell density of 1 × 10^5^ cells/mL. The mixture was transferred to a 6-well plate (3.94 mL/well) and incubated at 37 °C for cell culture. To feed the hiPSCs, 2–3 mL of complete mTeSR1 were added to the 6-well plate on day 4. Cell spheroids were harvested on day 5 and disassociated into the cell suspension for further analysis. The complete mTeSR1 medium was prepared by adding mTeSR1 supplements and PGgrow at a ratio of 1000:1 (*v*/*v*, medium: PGgrow).

### 2.4. hiPSC 3D Culture Medium Collection

For the 3D embedded culture, about 1.5 mL of the sample culture medium was gently aspirated above the cell–gel matrix without disturbing the hydrogel at day 3, 4, and 5. The collected medium was centrifuged at 3000× *g* for 10 min at 4 °C to remove any dead cells and debris. For the 3D suspension culture, the 3D suspension culture system was mechanically disrupted by pipetting on day 5. Then, the cell–gel mixture was transferred to a 15 mL conical centrifuge tube, and centrifuged at 700× *g* for 5 min with a swing bucket centrifuge. The supernatant was collected and filtered through a 0.2 μm filter to remove any gel fibers and to obtain medium samples. All collected media samples from the different culture conditions were stored at −20 °C for future use for either mass spectrum analysis or extracellular vesicles isolation.

### 2.5. Retrieving hiPSCs from the 3D Culture 

For the 3D embedded culture, the hydrogel was disrupted by pipetting (6–8 times) the gel and medium on top thoroughly. The mixture was then transferred to a conical tube and 20-fold of DPBS without calcium was added to dilute the solution for further separation. After centrifugation at 300× *g* for 5 min, the supernatant was discarded and the hiPSC spheroid pellets were collected from the bottom of tube. For passage or cryopreservation, the spheroids were then dissociated using TrypLE™ Express Enzyme (1X) (Thermal Scientific Fisher, Waltham, MA, USA) (1 mL/well for a 24-well plate) at 37 °C for 15 min to obtain single- or small-cluster hiPSC cells. Once the cells observed under microscope showed desirable results, the culture medium was added to stop the trypsinization. The cells can be either singularized into single cells for passage, further analysis, and cryopreservation or remained in spheroids for other direct uses such as drug testing or somatic cell differentiation, etc. 

For the 3D suspension culture, the hydrogel was mechanically disrupted by pipetting (6–8 times) and the mixture was transferred to a 50 mL conical tube. Then, the mixture was centrifuged at 700× *g* for 5 min by using a swing bucket centrifuge. The supernatant was removed before medium collection and the hiPSC spheroid pallets were collected for further use. For passage or cryopreservation, the hiPSC spheroids were dissociated into single or small cluster cells by using TrypLE Express Enzyme (1X) and incubated at 37 °C for 15–20 min. About 7 mL of TrypLE (1X) was used for one well of a 6-well plate. Once the cells observed under a microscope showed desirable results, the culture medium was added to stop the trypsinization.

### 2.6. hiPSC 2D Cell Culture

A 2D cell culture of hiPSC-F cells was employed as a control. To coat the 6-well plate, 0.5 mg of vitronectin (ThermoFisher) was diluted 100-fold to achieve a final concentration of 0.5 µg/cm^2^. The coated plate was either incubated in the incubator for 1 h or sealed with parafilm and stored at 4 °C for use within 7 days. A total of 1 × 10^5^ hiPSC-F cells were seeded onto the plate using mTeSR1 as the culture medium. The medium was replaced every day, and once the cell confluence reached 70% as observed under the microscope, the cells were harvested and passaged. 

To harvest the cells, the culture medium was removed, and the cells were washed with DPBS. Then, 1 × TrypLE was used to detach the cells from the plate. After adding 1mL TrypLE, the cells were incubated at 37 °C for 3–5 min. Once the cells observed under a microscope had a round shape and detached from the plate, 1 mL of culture medium was added to stop the trypsinization. The cell mixture was transferred to a 15 mL conical tube and centrifuged at 200× *g* for 5 min. The supernatant was removed to obtain the cell pellet for further cell counting and passage. 

### 2.7. Cell Count and Viability Measurement

After retrieving the cells following the procedure from Section 2.5, the Auto2000 Cellometer (Nexcelom Bioscience LLC, Lawrence, MA, USA) was used to determine cell number and viability using an acridine orange/propidium iodide (AO/PI) assay from Nexcelom Bioscience. The 20 µL well-suspended cell solutions were gently aspirated and then mixed with the 20 µL of AO/PI reagents. Both bright field and dual-fluorescence imaging were programed by the cellometer to detect live cells and dead cells. The live cells presented with the expression of fluorescent green and the apoptotic cells with early membrane damage were stained as fluorescent orange. Individual cells were counted for amount and diameter, then the data were automatically computed and reported for viability and concentration. The fold change was determined by dividing the total amount of harvested cells by the initial seeding density.

### 2.8. Expression of Pluripotent Biomarkers via RT-qPCR

Total RNA was extracted from the cells of 12 conditions using the TRIzol LS Reagent (ThermoFisher Sci.) and Direct-zol RNA MiniPrep kit (Zymo Research Corp, Irvine, CA, USA), and 100 µL of 1 × 10^6^/mL cells were used for each RNA extraction. The RNA concentration was measured by a NanoDrop spectrophotometer (ThermoFisher Sci.), and then normalized to 5 ng/µL for real-time qPCR reactions. The RT-qPCR was performed using the iTaq™ Universal SYBR Green One-Step kit (Bio-Rad, Hercules, CA, USA) on the CFX96 Touch Real-Time PCR Detection System (Bio-Rad) with the following program: initial reverse transcription at 50 °C for 10 min, denaturation at 95 °C for 1 min, followed by 40 cycles of amplification at 95 °C for 10 s, annealing at 60 °C or 52 °C, depending on the primer annealing temperature, for 30 s, and extension at 72 °C for 20 s. Nine primer sets (Table 2), targeting six pluripotent markers SOX2, OCT4, REX1, NANOG, UTF1, and hTERT, and three housekeeping genes hEID2, hCAPN10, and hZNF324B, were used in triplicate for each target in each run [11]. The average Ct value of the three housekeeping genes was used for gene expression level calculations for the six marker genes using the delta–delta Ct method [24]. 

### 2.9. Extracellular Vesicle Extraction

The supernatant was collected from the culture medium, then it was centrifuged at 3000× *g* for 15 min at 4 °C to remove dead cells and debris. The resulting supernatant was then subjected to further centrifugation at 10,000× *g* for 20 min at 4 °C to eliminate any remaining cell debris and microvesicles. The supernatant was subsequently transferred to an Amicon Ultra 5 mL centrifugal filter with a 3 kDa membrane (Merck Millipore, Burlington, MA, USA), which had been prewashed with DPBS and centrifuged at 14,000× *g* for 10 min. The Amicon filtration system was then centrifuged at 14,000× *g* for 30 min to concentrate the EVs and remove the peptides in the solution. Finally, the filter was inverted in a clean microcentrifuge tube and centrifuged at 4000× *g* for 10 min to collect the purified EVs.

### 2.10. EV Detection via NTA

To visualize and evaluate the size and concentration of the extracted EVs, nanoparticle tracking analysis (NTA) (NS500, NanoSight Ltd., Salisbury, UK) was applied. Before injection into the compartment, the EV solution underwent filtration using 0.22 µm syringe filters (VWR) to singularize the particles. Simultaneously, a non-conditioned control medium underwent the same process. A monochromatic laser beam was refracted at a low angle by the crystal apparatus before being introduced to the sample, which illuminated the nanoparticles captured by a conventional optical microscope. Five 30 s videos capturing the Brownian motion of individual particles were recorded and analyzed. These videos were subsequently analyzed utilizing the NTA software 2.3 (NanoSight Ltd.), which identified and tracked the Brownian movement of each particle, frame by frame. The two-dimensional Stokes Einstein equation was applied to calculate the particle size based on the velocity of particle movement [25]. The obtained results were presented in the form of a frequency-sized distribution graph, which depicted the number of particles and their corresponding diameters as computed from the video data. Subsequently, the concentration of released EVs was calculated to determine the average number of EVs, while the diameter data were used to specifically narrow down the nanoparticle population to specific EVs, such as exosomes.

### 2.11. Mass Spectrum (MS) for Protein Release Measurement 

Protein concentration was determined by BCA assay. Protein digestion was prepared using the S-Trap™ column (ProtiFi, Farmingdale, NY, USA) following the published protocol [26,27]. The digested sample contained a large excess of peptides originating from the proteins already present in the sample medium, such as albumin from mTeSR1 and transferrin in both the mTeSR1 and E8. Mass spectrometry was conducted under a data-independent acquisition (DIA) mode to achieve comprehensive coverage of low-abundant peptides. The DIA MS was performed under the MS^E^ mode as described in a prior study [27] using a Xevo G2-XS Q-Tof mass spectrometer (Waters) coupled with nano-ESI source and ACQUITY M-Class UPLC (Waters). Mass spectrum data analysis was performed using Progenesis QI for Proteomics (Waters). For ion accounting identification, the low-energy and elevated-energy settings were configured at 250 counts and 150 counts, respectively. For peptide identification, the following criteria were employed while using UniProt proteome (ID: UP000005640): fixed cysteine carbamidomethylation, variable methionine oxidation, allowance for up to 2 missed cleavages using trypsin, fragments/peptides ≥ 1, fragments/protein ≥ 3, peptides/protein ≥ 1, and a sequencing length of at least 6 amino acids. Non-conflicting peptides were used for protein quantitation.

### 2.12. Functional Enrichment Analysis

The protein profile acquired from proteomics was analyzed for the functional enrichment. The analysis was performed on up- and downregulated proteins in samples cultured in mTeSR1 and E8 media. “Scale” and “cluster” were applied to generate the heatmap for the protein profile. Uniport was utilized to screen the protein accession number within the review. Gene ontology (GO) biological functions enriched on differential protein sets were analyzed based on the gene ontology (GO) terms (http://www.geneontology.org/ (accessed on 11 November 2023)). The identified biological themes were sorted in the Database for Annotation, Visualization and Integrated Discovery (DAVID) terms https://david.ncifcrf.gov/home.jsp (accessed on 11 November 2023).

### 2.13. Statistical Analysis

GraphPad Prism was utilized for all statistical analyses, and the results are shown as mean with SEM. An unpaired two-tailed Student’s *t*-test and one-way ANOVA with Bonferroni’s multiple comparisons were used for evaluating significance. A significance level of *p* ≤ 0.05 denoted statistical significance (*, *p*  <  0.05; **, *p*  <  0.01; ***, *p*  <  0.001).

## 3. Results

### 3.1. hiPSC Proliferation Supported by the PGmatrix hiPSC 3D Culture System

The PGmatrix hiPSC for the 3D embedded culture was previously examined and compared with the 2D culture system, including Matrigel and vitronectin, leading to the observation of reduced variance in fold expansion and cell viability in 3D across multiple passages [11]. The PGmatrix hiPSC provided a stable growth environment compared to the 2D, and superior hiPSC properties compared to those of the existing advanced 3D technologies such as PEG-based hydrogels (Mebiol gel) and aggregated hiPSC spheroids generated by non-adherent U-Bottom 96-well plate [11]. Thus, PGmatrix hiPSC holds the potential for further optimization or adaptation to suit various cell culture conditions. In this context, several factors were considered, including the culture medium, hydrogel concentration and culture duration, and the choice between 3D embedded and 3D suspension PGmatrix systems. Within this framework, two specific hiPSC cell lines, one derived from fibroblasts (hiPSC-F) and the other from PBMC (hiPSC-P), were investigated within the PGmatrix hiPSC 3D culture systems (PGmatrix hiPSC 3D embedded (PG) and PGmatrix 3D suspension hiPSC (PG Susp)). Notably, the results demonstrated consistently high viability of hiPSCs, ranging from 95% to 99%, among all tested culture conditions (Figure 1A–D). However, the rate of cell proliferation was found to be culture condition-dependent, exhibiting significant variation in expansion fold, with values ranging from 7 to 20 (Figure 1A–D).

#### 3.1.1. hiPSC Lines

Both hiPSC lines, hiPSC-F and hiPSC-P, displayed similar growth performance within the PGmatrix hiPSC culture system (Figure 1A). For the E8 culture medium, a 0.3% gel concentration was selected, as the gel became stronger and the retrieval of spheroids became more challenging at a 0.5% concentration [11]. Under the same seeding density of 1 × 10^5^ cells/mL and cultured with mTeSR1 medium, both cell lines formed physiological 3D colonies (spheroids) with sizes ranging from 30 to 60 μM (Figure 1E–H). Notably, both cell lines exhibited significantly higher proliferation rates in the mTeSR1 medium, with a fold expansion ranging from 14 to 16 retrieved on day 5, compared to rates in the E8 medium, where the fold expansion was in the range of 9–11. Additionally, hiPSCs cultured in 0.3% PGmatrix with E8 showed reduced sphericity and a more polarized morphology (Figure 1G,H). The decreased cell growth rate and reduced sphericity within the E8 medium may be attributed to the lack of growth factors in E8 medium during cell growth [11]. 

#### 3.1.2. PGmatrix Concentration

Figure 1B demonstrates that high hydrogel concentrations of 0.5–1% provide better support for cell growth compared to the lower hydrogel concentration of 0.2%. As reported by [11], the 0.2% PGmatrix hiPSC exhibits an elastic modulus of 240 Pa, while the 0.5% PGmatrix shows an elastic modulus of over 600 Pa. The weaker gel at a 0.2% concentration can be easily disrupted during medium replacement on day 3 and day 4, leading to cell loss and reduced harvested-cell numbers. Moreover, the polarized cells observed within the 1% PGmatrix hiPSC with mTeSR1 (Figure 1J) further indicate that gel strength may exert an influence on the behavior of three-dimensional (3D) cell growth, particularly in the context of cellular morphology. This stress stimulation has the potential to stimulate induced pluripotent stem cells (iPSCs) and enhance various facets of cellular development, including derivation and maturation processes, such as for the generation of cardiomyocytes [28]. However, the PGmatrix gel strength at a 1% gel concentration was only about 1000 Pa, which should not be strong enough to alter hiPSC morphological shapes because the 3D hiPSC cultured in a modified PGmatrix at a 3% gel concentration with a gel strength of about 3000 Pa still presented a spherical shape [11]. A possible reason could be the smaller pore size of PGmatrix hiPSC at a 1% gel concentration due to the nature of its self-assembled nanofiber networks [20]. 

#### 3.1.3. Culture Duration

As depicted in Figure 1C, with a seeding density of 1 × 10^5^ cells/mL, hiPSCs exhibited a proliferation rate of approximately 7–8 folds at day 4, followed by a transition into the exponential growth phase with cell fold expansion reaching 14–16 at day 5. Subsequently, the cell growth rate appeared to decelerate, and the cell population proliferated further to 17–19 folds by day 6. Based on the data, the optimal culture duration for hiPSCs appears to be at day 5, during the exponential growth phase, which is conducive for retrieving the cell spheroids for various downstream hiPSC-based applications. The exponential growth phase may vary with seeding density, cell type, or culture conditions (i.e., medium type and hydrogel conditions). 

#### 3.1.4. Three-Dimensional Embedded vs. Three-Dimensional Suspension Cultures

Figure 1D presents the comparison of 3D hiPSC-P growth performance cultured in 3D embedded (PG) and 3D suspension (Susp) conditions with mTeSR1 medium. The results indicated that the hiPSCs in PGmatrix 3D Susp exhibited similar hiPSC-P cell proliferation with the embedded system, with 15–19 folds (Figure 1D P-mT-Susp). A modified PGmatrix 3D Suspension (MSusp) with higher viscosity was used for 3D hiPSC-P culture (P-mT-MSusp) with the hope of increasing cell holding capacity. Cells in MSusp presented similar viability as those in Susp (Figure 1D); however, such a high viscosity resulted in a higher cell loss rate in the process of retrieving cells, leading to lower proliferation folds. 

### 3.2. Stable Pluripotent Gene Expression of hiPSCs Cultured in 3D PGmatrix hiPSC Systems 

To evaluate the genetic effects under five primary culture conditions, the expression levels of six genes (*SOX2*, *OCT4*, *REX1*, *NANOG*, *UTF1*, and *hTERT*) associated with pluripotency stem cells were quantitatively determined by real-time qPCR in hiPSC-P. The hiPSC-F cultured in PGmatrix hiPSC at a 0.5% gel concentration with mTeSR1 complete medium was used as a control, which was reported to show superior or similar gene expression compared to hiPSCs cultured in 2D [11]. According to our preliminary testing, three embryonic genes, *hEID2*, *hCAPN10*, and *hZNF324B*, were used as housekeeping genes. Overall, all pluripotent genes remained relatively stable, with minor variation among all the culturing conditions, typically ranging from one to two-fold changes (Figure 2A–D). 

Comparison of hiPSC-P and hiPSC-F cultured in PGmatrix hiPSC at a 0.5% gel concentration with mTeSR1 showed similar gene expression levels as hiPSC-F (Figure 2A). Pluripotent gene expression analysis of hiPSC-P cultured in mTeSR1 revealed better the adaptation of cells (Figure 2A). In contrast, for hiPSC-F, similar gene expressions were observed in both the mTeSR1 and E8 media, showing better adaption to the E8 medium than hiPSC-P (Figure 2A). The better adaptation of hiPSC-F to the culture medium might be attributed to their epigenetic memory as adhesive cells. Despite being induced into iPSCs from parental cells, not all epigenomic features may be fully cleared even after extensive passaging, potentially influencing cellular behavior and responses to the culture environment. Conversely, hiPSC-P performed better in the mTeSR system over the E8 (Figure 2A) and in the Susp system over the embedded system (Figure 2D), with increased expression in SOX2 and REX1, and decreased expression in hTERT. As PBMCs mainly consist of suspension cells, their superior performance in suspension culture aligns with their physiological characteristics. In addition, culture duration from day 4 through day 6 and hydrogel strength from 0.2 to 1% showed no significant effects on pluripotent gene expression (Figure 2B,C). 

### 3.3. Protein and Genetic Profile Secreted by hiPSCs 

Mass spectrometry-based analysis of secretome was conducted to investigate the differences in biological enrichments secreted by hiPSCs cultured in various PGmatrix culture conditions. Specifically, hiPSC-F cultured in 2D on vitronectin and 3D 0.5% PGmatrix hiPSC as well as hiPSC-P cultured in 0.5% PGmatrix hiPSC were examined in the mTeSR-based culture medium. Additionally, the secretome of hiPSC-F and hiPSC-P cultured in the 3D system in the E8 medium at 0.3% PGmatrix hiPSC were also analyzed. The predominant protein constituents within the culture medium are associated with essential cellular components, signaling molecules, metabolic proteins, and functional enrichments. Among a total of 757 proteins detected in the mTeSR-based culture medium, 200 proteins were identified confidently, as they contained three or more unique peptides. Similarly, among 641 proteins detected in the E8-based culture medium, 210 proteins were identified confidently. The MS results revealed the presence of exosome-related proteins, including heat-shock proteins such as HSP 90-alpha (HSP90AA1), antigen protein MHC class I related protein (MICA), integrins alpha-M (ITGAM), and cell surface A33 antigen (GPA33, an immunoglobulin family member). It is noteworthy that GPA33 had been reported to be present in the collected medium sample previously [29]. The mass spectrum (MS) results revealed a cohort of proteins intricately involved in angiogenesis signaling pathways, which includes vascular endothelial growth factor (VEGF), bone morphogenetic proteins-2- inducible protein kinase (BMP2K), and thrombospondin-related proteins, contributing to multicellular organism development (Appendix A). Specific proteins were linked to vital biological processes, such as wound healing, differentiation of cells, organ formation, and anatomic structures, as exemplified by the presence of epidermal growth factors (EGF) (Appendix A) [30]. The findings showed the presence of proteins that stimulate cardiomyogenesis, such as fibroblast growth factors (FGFs), which also demonstrated insulin-like growth factors (IGF), exhibiting the promotion of proliferation in cardiac, endothelial, and smooth muscle cells [31]. The discovery of surface receptors such as EGF and signaling molecules including Rhos and mitogen-activated protein kinase (MAPK), that are responsible for governing self-renewal and cell differentiation, further emphasizes the clinical significance of these protein constituents [31]. These findings provide valuable insights into the secretome of hiPSC under different culture conditions, shedding light on the molecular components and signaling pathways that may play crucial roles in cellular behavior and interactions within the 3D PGmatrix system. The functional enrichments detected from these conditions have been found to promote cell survival, proliferation, and differentiation, as well as facilitate wound healing and tissue repair.

#### 3.3.1. Protein Profile Analysis

Protein regulation is depicted through the heatmaps shown in Figure 3A,B, representing four mTeSR1-treated and two E8-treated conditions, respectively. The heatmaps provide visual representations of the discernible disparities among proteins excreted by the hiPSCs following 5-day incubation under various conditions. For the 3D embedded culture conditions, hiPSC-P excreted slightly more proteins than hiPSC-F in both the mTeSR1 and E8 media (Figure 3A,B), and both excreted more proteins in 3D conditions than in 2D (Figure 3A). Conversely, hiPSCs in the 3D suspension matrix presented a prevalent trend of more upregulated proteins than those in the 3D embedded conditions (Figure 3A). Unique protein profiles were observed for the 3D suspension matrix system in contrast to the 3D embedded ones (Appendix A), likely due to the influences of the culture system or cell–gel matrix separation methods. Specifically, medium samples collected from the 3D suspension matrix were cultured with 4 × 10^5^ cells as a seeding density, while the embedded system was cultured with a seeding density of 1 × 10^5^ cells. With the amount of downregulation in each condition, the F-mT-2D condition show fewer downregulated proteins than the 3D conditions, indicating that with the 3D culture systems, cells are more able to produce the nutritious proteins from the medium and secrete more enrichments. The color scheme patterns show different levels of cell activities at the protein level. This finding suggests that iPSCs within a 3D PGmatrix hiPSC environment secrete more unique proteins that cannot be observed in a 2D culture setting. 

#### 3.3.2. Functional Gene Ontology Analysis

Gene ontology (GO) analysis was performed in this study to assess the proteome signature associated with the differentially expressed genes under the mTeSR1 and E8 culture systems. The results revealed the presence of both congruities and disparities in the biological functional enrichments of hiPSCs between the two culture media mTeSR1 and E8 (Figure 3C,D, Appendix A). The enrichment for the biological process (BP) category was relatively low (Appendix A). Interestingly, the mTeSR1-based system exhibited more enrichments in the cellular component (CC) category than the E8-based system, but was similar in the molecular function (MF) category (Appendix A). Notably, the most abundant proteins excreted from hiPSCs in both the mTeSR1 and E8-based systems were found to be involved in the protein binding (GO:0005515) in the category of molecular function (MF), with 141 target genes for mTeSR1 and 135 target genes for E8 (Figure 3C,D, Appendix A). Additionally, other enriched proteins in the count between mTeSR1 and E8, such as metal ion binding (GO:0046872), DNA binding (GO:0003677), and RNA binding (GO:0003723) are also similar (Figure 3C,D). For molecular function, the mTeSR1-based systems exhibited higher enrichment in protein activities, while the E8-based systems showed enriched proteins related to binding regulation (Figure 3C,D), which align with the different purposes and application of these culture media. The E8 is usually a good selection during the cell reprogramming process, while mTeSR1 is suitable for the iPSC maintenance. Based on the GO analysis, the critical functions of the proteins were predicted. For example, proteins protecting cells against oxidative damage (cilia- and flagella-associated proteins) defined by the GO group (GO:0016209) were identified by MS to be involved in signal transduction, which also contributes to cell motility. In addition, proteins implicated in EV biogenesis from the previous research were characterized and in agreement with our analysis [32], These proteins were found to influence many cell functions, not limited to cell proliferation, iPSC differentiation, cell survival, and angiogenesis. Another example is the protein responsible for intracellular trafficking and fusion, such as the RAB-related protein (Appendix A). These findings highlight the distinct biological processes and functional enrichments produced by the differences in hiPSC-P behaviors under different culture conditions. 

#### 3.3.3. Release of Extracellular Vesicles

hiPSCs have been used as a tool to explore extracellular vesicles (EVs) that carry therapeutic peptides or nucleic sequences within their membranes [33]. The EVs derived from hiPSCs have demonstrated their potential as potent mediators of essential cellular processes, which might share similar functions as hiPSCs themselves. These EVs have shown their capacity not only to promote cell survival and proliferation but also to orchestrate cellular differentiation. Furthermore, EVs facilitate wound healing and tissue repair, which shows their significance in regenerative medicine [32]. The nanoparticle tracking analysis (NTA) performed on mTeSR1-based culture systems revealed the presence of components related to extracellular exosomes (Figure 4, Table 3). The result was supported by the detection of proteins (GO: 0070062-extracellular exosome) related to exosomes in the MS analysis (Figure 3C, arrow). Significantly, the enriched angiogenic proteins identified through MS analysis have been found within the EVs derived from iPSCs, encompassing crucial factors such as VEGF, EGF, and IGF [34,35], of which the precise localization and functional significance were studied [30]. The distinct patterns of EVs secreted by hiPSCs across various culture systems were also revealed (Figure 4B,D). Notably, the EV particles secreted by different culture systems highlighted substantial disparities in EV secretion between the 3D (2 × 10^8^) and 2D (1 × 10^8^) systems (Figure 4A). Specifically, the 3D culture system demonstrated a significantly higher rate of EV secretion than the 2D system, although the best culture condition recommended by the manufacturer was used for 2D, implying that the 3D environment exerts a crucial influence on promoting the release of EVs. Within the E8-based culture systems, hiPSC-F (2 × 10^8^) exhibited greater EV secretion than hiPSC-P (9 × 10^7^) (Figure 4A,C). However, hiPSC-P cultured in 3D 0.5% PGmatrix systems with the mTeSR medium demonstrated higher EV secretion (1.67 × 10^8^) that that in the E8 medium (9 × 10^7^) (Figure 4A). 

#### 3.3.4. Growth Factors and Extracellular Matrix (ECM) Proteins

Among all the proteins identified by MS (at detectable threshold set), hiPSCs cultured with the mTeSR1 (mT) medium released elevated levels of a variety of growth factors, whereas those cultured with the E8 medium secreted more diverse ECM proteins (Appendix A). The hiPSC-P with mT in the PGmatrix 3D suspension (PG 3D Susp) matrix showed significantly higher levels of Semaphorin 7A (Sema7A), serotransferrin (TF), and insulin growth factor binding protein 2 (IGFBP2) than those in the 2D cultures (Figure 5A) (*p* < 0.001, *p* < 0.001 and *p* = 0.018, respectively). Conversely, hiPSC-F and hiPSC-P with mT in the 0.5% PGmatrix 3D (PG 3D) embedded system released similar levels of Sema7A, TF, and IGFBP2, in which IGFBP2 was elevated and Sema7A and TF were reduced, compared to results in 2D (Figure 5A) (*p* < 0.001, *p* < 0.001 and *p* = 0.018, respectively). Sema7A is known mostly for its role in T-cell and neuronal development [36,37]. However, Sema7A can also regulate actin reorganization and integrin binding through plexins [38,39]. In the 0.5% PG-3D embedded system, the gel strength was able to hold and support hiPSC spheroid conformation during growth, while such supportive force was much weaker in the PG-3D suspension, and cells may need to re-organize their cytoskeletons as they proliferate. Conversely, in the 2D culture, hiPSCs are firmly attached to the ECM-coated surface with minimal need to adjust integrin binding as they expand their colonies. Thus, we observed 2D secretome having a similar level of Sema7A as the mTeSR1 fresh (blank) while hiPSC-P with mTeSR1 in PG 3D Susp generated significantly more Sema7A proteins. The TF, on the other hand, can serve as a growth factor [40,41] in addition to its roles as an iron transporter and antioxidant [42]. In both 2D and PG 3D Susp, hiPSCs had more contact with the medium and indirectly with the atmosphere, so they may need more protection against oxidative stress compared to those in the 0.5% PG 3D embedded system. A stronger gel with dense peptide nanofibers would reduce the oxygen content in the culture, which generates a hypoxic environment, hence the elevated level of TF observed in the 0.5% PG 3D embedded matrix, compared to the mTeSR1 blank, could act more as growth factor for hiPSC maintenance. Another important protein is IGFBP2, which is a regulator protein of insulin growth factors (IGFs) and exists in highly proliferative tissues [43]. IGFBP2 was reported to support the growth of various stem cells including neural stem cells and hematopoietic stem cells [44,45]. IGFBP2 is one of the key proteins in regulating hiPSC proliferation and may be of higher importance for the 3D culture, which supports the observations in this study. 

On the other hand, the hiPSC secretome in the E8 medium largely did not overlap with the secretome in the mTeSR1. The hiPSC-P and hiPSC-F secretome in the E8 medium released significantly higher amounts of albumin (ALB), collagen alpha-1(XIV) chain (COL14A1), and collagen alpha-1(I) chain (COL1A1), fibroblast growth factor 17 (FGF17). and latent-transforming growth factor beta-binding protein 4 (LTBP4) than in the blank fresh E8 medium (Figure 5B). Albumin is not present in the E8 blank medium because it has been reported that bovine serum albumin (BSA) had no effect on hiPSC growth in a 2D culture [46]. Our results suggested that albumin may be important for the 3D culture because the hiPSCs in the medium containing BSA (i.e., mTeSR1) released abundant growth factors and had better proliferation than those in the BSA-free medium (i.e., E8), despite being under the same 3D environments (Figure 1A). The COL14A1 and COL1A1 have been reported to promote the expansion of certain stem cells [47,48], they may also contribute to ECM matrix modification in a 3D culture [11]. FGF17, a member of the FGF8 subfamily, can affect proliferation by regulating various downstream signaling pathways including Ras/Raf-MEK-MAPKs (mitogen-activated protein kinases), and phosphatidylinositol-3 kinase/protein kinase B (PI3K/AKT) [49]. FGF8 was also found elevated in the hiPSC secretome in E8 (Appendix A). Both FGF17 and FGF8 play critical roles in the development of various organs [49], but FGF17 in particular has been reported to activate progenitor cell proliferation to boost rejuvenation [50]. LTBP4, LTBP3, and LTBP1 were all identified in the hiPSC secretome in E8, but LTBP4 was identified with the highest confidence (10 unique peptides) (Appendix A). LTBPs have structural similarity to fibrillin and can be incorporated into the ECM matrix. They also serve as key components of transforming growth factor beta (TGFβ) signaling and thus participate in hiPSC maintenance [51,52]. LTBP4, among other LTBPs, has been shown to modulate elastic fiber assembly [53], which may help hiPSCs to adapt their environment during growth [11]. These observations suggested that, when in the E8 medium, 3D-cultured hiPSCs put in more effort to modulate and reorganize the surrounding matrix by releasing the needed components for sustaining living conditions. However, when in mTeSR1, the 3D-cultured hiPSCs secreted more growth-regulating factors, which led to the growth performance reported in Figure 1A. These findings demonstrated that PGmatrix hiPSC is a suitable 3D culture system for hiPSCs, allowing modulation of medium composition to achieve a desirable quality of either secretome substances or hiPSC cell performance for downstream applications.

## 4. Discussion

The 3D PGmatrix hiPSC system has demonstrated significant advantages over traditional 2D culture methodologies, notably in fostering sustained cell proliferation across multiple passages, as evidenced by recent studies [11]. This robust cell proliferation is complemented by the unwavering and stable expression of pluripotency-associated genes, as depicted in Figure 2A–D. These findings firmly establish the 3D PGmatrix hiPSC culture system as a dependable platform for maintaining hiPSCs in a pluripotent state. Notably, the hydrogel matrix exhibits the requisite mechanical strength, features unique deformability and recombination capabilities [18], and meets the criteria for providing cells with a stable culture system. This adaptability extends to encompass a diverse range of hydrogel concentrations, culture medium selection and supplement additions, underscoring its versatility for accommodating various cell types and diverse research applications, especially for the hiPSC-induced differentiation model, hiPSC maintenance, and hiPSC EVs production. 

Beyond its pronounced cell proliferation benefits, the PGmatrix hiPSC system with peptide scaffold networks [11], offers substantial advantages in terms of protein adsorption, secretion dynamics, and cellular enrichment release. The 3D culture system serves as a critical bridge between in vitro and in vivo chemical drug screening, furnishing a more realistic and reliable platform for evaluating the efficacy of potential drugs or compounds. Cells embedded in the 3D system better mimic the in vivo environment, and better demonstrate the efficiency and effectiveness for disease models and drug screening. Notably, researchers can observe the effects of drugs on head and neck cancer cell proliferation, migration, and invasion more authentically [19]. Furthermore, previous research has shown that the interaction between the hydrogel and hydrophobic drugs is negligible, facilitating effective drug diffusion within the hydrogel and action on the cells [20]. The tunable pore dimensions within the 3D PGmatrix hydrogel scaffold significantly enhance nutrient transport, thereby enabling precise control over drug delivery. 

The 3D PGmatrix hiPSC system also benefits and facilitates directed cell differentiation, especially for liver, cardiac, and neurosomatic cells, which were known for long maturation times and difficulty in maintenance [54]. The EVs derived from iPSC-derived cardiomyocytes (iCMs) have shown the ability to promote angiogenesis and enhance the expression of dystrophin for Duchenne muscular dystrophy [55]. With EVs’ protective effects in various conditions, the application extends to skin aging, liver fibrosis, cardiovascular diseases, and neurological disorders, regulation in apoptosis, inflammation, fibrosis, and angiogenesis, and delivery of various cargoes including miRNAs, small molecules, and proteins to target cells [32]. The 3D PGmatrix cell culture promotes the secretion of in vivo like extracellular vesicles (EVs) [12]. The distinct PGmatrix hydrogel structure uniquely positions it for EV research, providing a precisely regulated microenvironment conducive to the study of EV secretion dynamics and their intricate interactions with encapsulated cells. The EVs derived from hiPSCs within the 3D embedded PGmatrix hiPSC system (Figure 4) can be considered a potent co-product of hiPSC production for therapeutic research and applications. 

In addition, hiPSCs cultured in 3D PGmatrix hiPSC systems (regardless whether 3D embedded or suspension) secreted either more growth factors (Figure 5A) or more ECM proteins (Figure 5B) that establish desirable living environments for themselves according to a given culture condition. These compelling findings collectively underscore the practicality and efficacy of 3D PGmatrix hiPSC culture systems, positioning these systems as an instrumental asset for preserving hiPSC pluripotency and facilitating a wide range of research and therapeutic applications. Although concerns regarding the erasure of epigenetic memory through prolonged culture persist [11], 3D PGmatrix hiPSC culture systems exhibit remarkable adaptability to diverse cell types and specific research paradigms, promising sophisticated applications including organoid generation and cell differentiation. 

## 5. Conclusions

This study offers a multifaceted understanding of the intricate relationship between culture conditions and the behavior of human-induced pluripotent stem cells (hiPSCs). Culture conditions modulated hiPSC growth, gene expression, protein secretion, and extracellular vesicle (EV) release. The hiPSCs cultured in the 3D PGmatrix hiPSC system demonstrated the ability to maintain consistently high cell viability above 95% and expansion folds (10–20) depending on culture conditions, along with stable pluripotent gene expression. The hiPSCs in a 3D culture released significantly higher numbers of unique proteins and EVs than those in a 2D culture. In addition, hiPSCs in a 3D culture secreted desirable growth factors and ECM proteins to modify their living environment according to a given culture condition. These findings are underpinned by the unique porous PGmatrix hydrogel scaffold and cytocompatibility, which not only provides structural support but also facilitates the efficient transport of crucial proteins involved in cell secretion and allows for seamless supplement addition. The 3D PGmatrix hiPSC culture system holds a promising platform for hiPSC culture and downstream applications such as differentiation of somatic cells, tissue engineering via bioprinting, drug screening, cell therapies, and disease modeling. 

## Figures and Tables

**Figure 1 cells-13-00143-f001:**
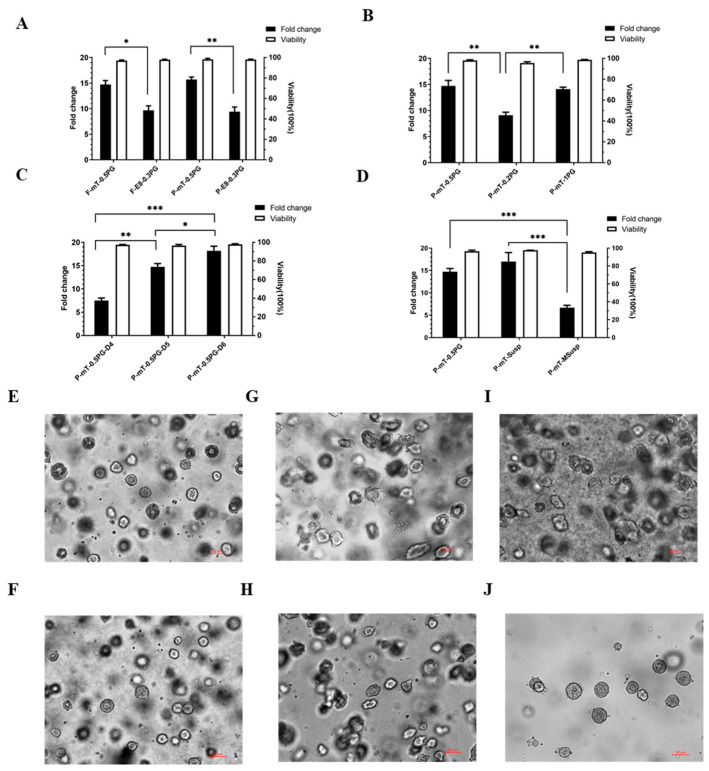
Effects of 3D culture conditions (PGmatrix hiPSC for 3D embedded (PG) and PGmatrix 3D suspension hiPSC (Susp)) and cell lines on hiPSC growth viability and proliferation. (**A**) Comparison of cell line and culture medium (passage 9–11): hiPSCs derived from fibroblast cells (hiPSC-F) (Applied Stemcell) vs. PBMC (hiPSC-P) (Applied Stemcell); culture medium mTeSR1 (Stemcell Technologies) vs. Essential 8™ Medium (ThermoFisher); (**B**) comparison of the 3D embedded PG hydrogel concentration (passage 9–11): 0.2% vs. 0.5% vs. 1% PG concentration; (**C**) comparison of 3D embedded culture duration (passage 9–11): 4 days’ culture vs. 5 days’ vs. 6 days’; (**D**) comparison of 3D PGmatrix hiPSC culture systems (passage 9–11): 3D embedded system (PG) vs. 3D suspension system (Susp) and modified suspension system (MSusp); (**E**) hiPSC-F in mTeSR1: cell morphology of hiPSC-F cultured in 3D 0.5%PG mTeSR system; (**F**) hiPSC-P in mTeSR1: cell morphology of hiPSC-P cultured in 3D 0.5%PG mTeSR1 system; (**G**) hiPSC-F in E8: cell morphology of hiPSC-F cultured in 3D 0.3%PG E8 system; (**H**) hiPSC-P in E8: cell morphology of hiPSC-P cultured in 3D 0.3%PG E8 system; (**I**) hiPSC-P in 1% hydrogel: cell morphology of hiPSC-P cultured in 3D 1%PG mTeSR1 system; (**J**) hiPSC-P in 3D suspension hydrogel: cell morphology of hiPSC-P cultured in 3D suspension mTeSR system. (All scar bars stand for 50 µm.). * *p* < 0.05, ** *p* < 0.01, *** *p* < 0.001.

**Figure 2 cells-13-00143-f002:**
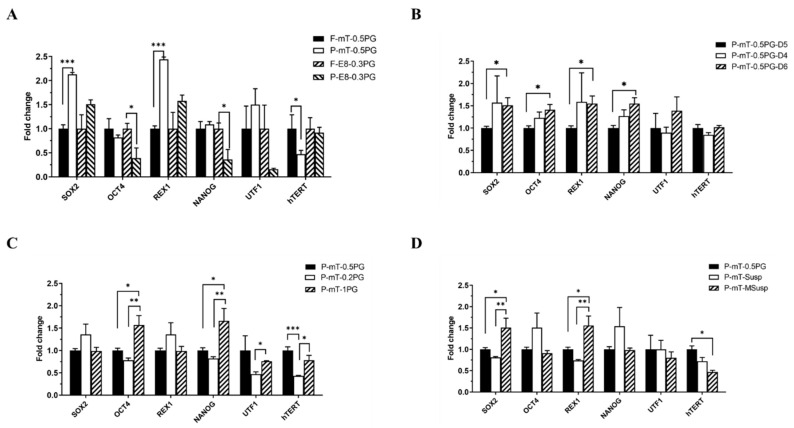
Effects of 3D culture conditions and cell lines on hiPSC pluripotent gene expressions. (**A**) Pluripotent gene expression of hiPSC-P vs. hiPSC-F cultured in 3D 0.5%PG mTeSR1 system, and hiPSC-P vs. hiPSC-F cultured in 3D 0.3%PG E8 system; (**B**) pluripotent gene expression of hiPSC-P cultured in 3D 0.5%PG mTeSR1 system for 4 days vs. 5 days vs. 6 days; (**C**) pluripotent gene expression of hiPSC-P cultured in 3D 0.2% vs. 0.5% vs. 1% PG mTeSR1 system; (**D**) pluripotent gene expression of hiPSC-P cultured in the 3D embedded system vs. the 3D suspension system and the modified-suspension system. * *p* < 0.05, ** *p* < 0.01, *** *p* < 0.001.

**Figure 3 cells-13-00143-f003:**
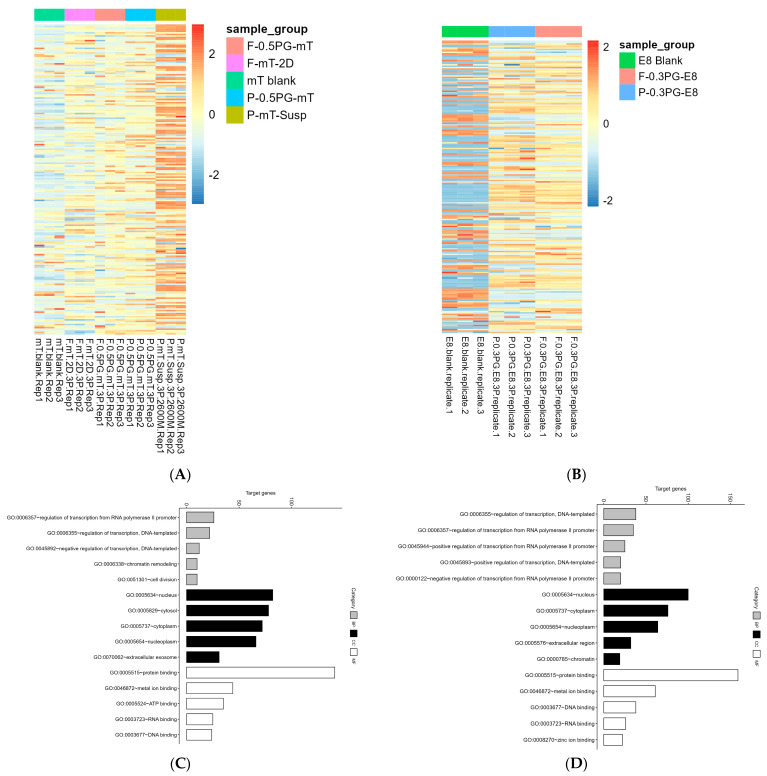
Proteomics analysis of hiPSC secretome release. (**A**) The protein profile analysis based on the mass spectrum (MS) analysis results of medium collected from hiPSC-P and hiPSC-F cultured in mTeSR1-based systems; warm colors indicate the increase in protein levels, while cool colors show the decrease; (**B**) the protein profile analysis based on the MS results of medium collected from hiPSC-P and hiPSC-F cultured in an E8-based system; (**C**) gene ontology (GO) analysis based on the combined MS results of media collected from hiPSC-P and hiPSC-F cultured in mTeSR-based systems, showing the target gene number and three categories, which are biological process (BP), molecular function (MF), and cellular compound (CC); (**D**) GO analysis based on the combined MS results of media collected from hiPSC-P and hiPSC-F cultured in an E8-based system, showing the target gene number and categories.

**Figure 4 cells-13-00143-f004:**
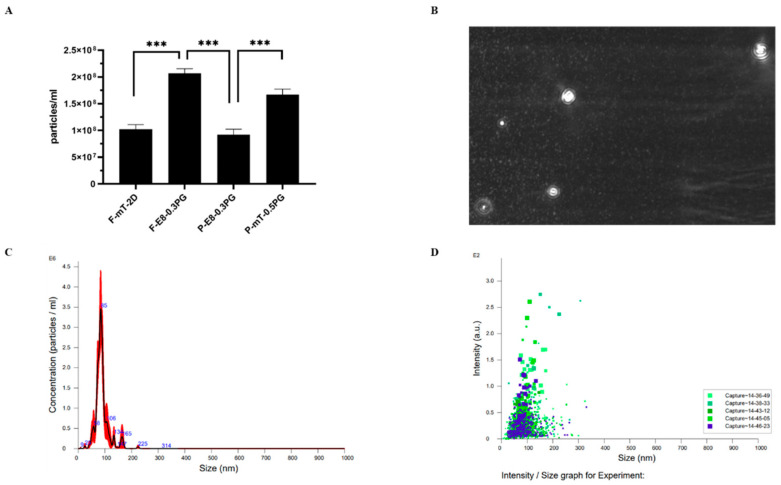
EV secretion of hiPSCs in varied culture conditions. (**A**) The release of extracellular vesicles (EVs) from hiPSCs cultured in different conditions; (**B**) the NTA detections of EVs via the dynamic light scattering method; (**C**) the computation results of five captures regarding the diameter and concentration of the EVs; (**D**) the intensity of EVs over size shown by computational scattering of all particles captured over five times. *** *p* < 0.001.

**Figure 5 cells-13-00143-f005:**
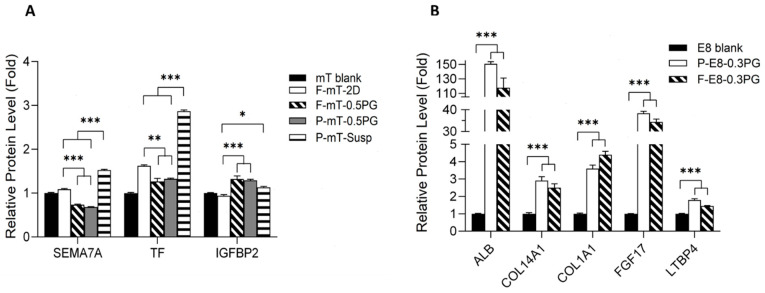
Levels of representative proteins detected in conditioned medium from different hiPSC cultures with the mTeSR1 medium (**A**) and E8 medium (**B**). Fold calculation based on mT blank and E8 blank. Data shown as mean ± standard error (SE). Comparison based on ANOVA, Tukey correction for post hoc tests. * *p* < 0.05, ** *p* < 0.01, *** *p* < 0.001.

**Table 1 cells-13-00143-t001:** Grouped culture conditions (short name for Figure labels).

hiPSC-hPBMC-mTeSR-0.5%PG concentration (P-mT-0.5PG)
hiPSC-fibroblast-mTeSR-0.5%PG concentration (F-mT-0.5PG)
hiPSC-hPBMC-E8-0.3%PG concentration (P-E8-0.3PG)
hiPSC-fibroblast-E8-0.3%PG concentration (F-E8-0.3PG)
hiPSC-hPBMC-mTeSR-0.5%PG concentration-Day4 (P-mT-0.5PG-D4)
hiPSC-hPBMC-mTeSR-0.5%PG concentration-Day5 (P-mT-0.5PG-D5)
hiPSC-hPBMC-mTeSR-0.5%PG concentration-Day6 (P-mT-0.5PG-D6)
hiPSC-hPBMC-mTeSR-0.2%PG concentration (P-mT-0.2PG)
hiPSC-hPBMC-mTeSR-1%PG concentration (P-mT-1PG)
hiPSC-hPBMC-mTeSR-Suspension (P-mT-Susp)
hiPSC-hPBMC-mTeSR-modified-Suspension (P-mT-MSusp)
hiPSC-fibroblast-mTeSR-2D (F-mT-2D)

**Table 2 cells-13-00143-t002:** Real time quantitative PCR (RT-qPCR) primers.

Gene	Primer ID	Primer Sequence	Tm (°C)
SOX2	SOX2-F1	CAACCAGAAAAACAGCCC	52
	SOX2-R1	TCTCCGACAAAAGTTTCC	
REX1	REX1-F	GTTTCGTGTGTCCCTTTC	52
	REX1-R	CTTTCCCTCTTGTTCATTC	
OCT4	OCT4-F	AAAGAGAAAGCGAACCAG	52
	OCT4-R	CCACATCCTTCTCGAGCC	
NANOG	NANOG-F1	TGTGATTTGTGGGCCTGA	52
	NANOG-R1	GTGGGTTGTTTGCCTTTG	
UTF1	UTF1-F	CTCCCAGCGAACCAG	52
	UTF1-R	GCGTCCGCAGACTTC	
hTERT	hTERT-F	GGAGCAAGTTGCAAAGCATTG	60
	hTERT-R	TCCCACGACGTAGTCCATGTT	
DNMT3B	DNMT3B-F	GGAGCCACGACGTAACAA	60
	DNMT3B-R	GGCATCCGTCATCTTTCA	
hEID2	hEID2-F	GAAGCCTGCAGAGCAAGG	60
	hEID2-R	ATATCGAGGTCCACCCTGTG	
hCAPN10	hCAP-F	GGAGGTGACCACAGATGACC	60
	hCAP-R	GTAAGGGGAGCCAGAACACA	
hZNF324B	hZNF-F	GAGAATGGCCACGAGCTTT	60
	hZNF-R	TTTACACTGTGGCAGGCATC	

**Table 3 cells-13-00143-t003:** Extracellular vesicle (EVs) particles by NTA analysis.

	Sample	Particles/mL	Mean Diameter	
1	hiPSC-F-2D-mTeSR	1.02 × 10^8^ ± 9.04 × 10^6^	100.00	nm
2	hiPSC-F-0.3%PG-E8	2.07 × 10^8^ ± 8.35 × 10^6^	88.00	nm
3	hiPSC-P-0.3%PG-E8	9.21 × 10^7^ ± 1.02 × 10^7^	101.1	nm
4	hiPSC-P-0.5%PG-mTeSR	1.67 × 10^8^ ± 1.02 × 10^7^	176.4	nm

## Data Availability

Data are contained within the article and Appendix A.

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
