# Peer review of "Unlocking the Potential of Human-Induced Pluripotent Stem Cells: Cellular Responses and Secretome Profiles in Peptide Hydrogel 3D Culture"

_cells, 2024, doi:10.3390/cells13020143_

Round 1

Reviewer 1 Report

Comments and Suggestions for Authors

This manuscript presents a comparative study of two hiPSC lines within 3D culture systems, utilizing the advanced PGmatrix-hiPSC technology. The authors observed the formation of hiPSC spheres in 3D culture, noting the maintenance of cell proliferation and pluripotency. Additionally, they characterized the secretome and extracellular vesicles during 3D culture. The manuscript was well-written and the presented results support their conclusions. I have a few suggestions for improvement:

1. The Methods section should include a description of the statistical analysis.

2. Clarify the process of collecting and digesting cells from 3D culture systems in "2.7. Cell count and viability measurement." Additionally, provide details on how to calculate the "fold change" depicted in Figure 1.

3. Include relevant cell state information, such as passage number and subculture timing, in the legend or result description of Figure 1 A-D. Consider adding subtitles for each panel in Figure 1 E-J to enhance readability.

4. Correct the subtitle numbering in the Results section to follow the correct order, such as 3.1, 3.2, etc

Comments on the Quality of English Language

The manuscript was well-written.

Reviewer 2 Report

Comments and Suggestions for Authors

iPSCs are progressively emerging as the primary reservoir of patient-specific and disease-specific cellular material used in in vitro disease modeling. In this manuscript, the authors detect the performance, integrity, and secretome of iPSCs under 3D culture in PGmatrix. Several questions require further consideration.

1.     In vitro, iPSCs have the capability to differentiate into target cells such as neurons, cardiomyocytes, blood cells, bone, etc. The authors should experimentally compare or discuss the differentiation potential of iPSCs under 3D culture conditions.

2.     iPSCs have the capability to form teratomas in vivo. The authors should conduct experimental comparisons or discussions concerning the in vivo teratoma formation ability of iPSCs under 3D culture conditions.

Comments on the Quality of English Language

The sentence structure and punctuation need refinement throughout the text. For instance, in the second paragraph of the discussion, the phrase "Cells embedded in the 3D system better mimics the in vivo environment, and better demonstrates the efficiency and effectiveness for disease models and drug screening" lacks proper punctuation. "Notably" should begin a new sentence. The authors should review the entire document for similar issues.
